# Anabasine and Anatabine Exposure Attributable to Cigarette Smoking: National Health and Nutrition Examination Survey (NHANES) 2013–2014

**DOI:** 10.3390/ijerph19159744

**Published:** 2022-08-08

**Authors:** Patrick B. Bendik, Sharyn M. Rutt, Brittany N. Pine, Connie S. Sosnoff, Benjamin C. Blount, Wanzhe Zhu, June Feng, Lanqing Wang

**Affiliations:** 1Tobacco and Volatiles Branch, Division of Laboratory Sciences, National Center for Environmental Health, Centers for Disease Control and Prevention, Atlanta, GA 30341, USA; 2Office of Science, Center for Tobacco Products, U.S. Food and Drug Administration, Silver Spring, MD 20992, USA

**Keywords:** tobacco biomarker, cigarette smoking, nicotine replacement therapy, cut point, anabasine, anatabine, cotinine, NHANES

## Abstract

Anabasine and anatabine are minor alkaloids in tobacco products and are precursors for tobacco-specific nitrosamines (TSNAs). The levels of these two compounds have been used to differentiate tobacco product sources, monitor compliance with smoking cessation programs, and for biomonitoring in TSNA-related studies. The concentrations of urinary anabasine and anatabine were measured in a representative sample of U.S. adults who smoked cigarettes (*N* = 770) during the 2013–2014 National Health and Nutrition Examination Survey (NHANES) study cycle, which was the first cycle where urinary anabasine and anatabine data became available. Weighted geometric means (GM) and geometric least squares means (LSM) with 95% confidence intervals were calculated for urinary anabasine and anatabine categorized by tobacco-use status [cigarettes per day (CPD) and smoking frequency] and demographic characteristics. Smoking ≥20 CPD was associated with 3.6× higher anabasine GM and 4.8× higher anatabine GM compared with smoking <10 CPD. Compared with non-daily smoking, daily smoking was associated with higher GMs for urinary anabasine (1.41 ng/mL vs. 6.28 ng/mL) and anatabine (1.62 ng/mL vs. 9.24 ng/mL). Urinary anabasine and anatabine concentrations exceeded the 2 ng/mL cut point in 86% and 91% of urine samples from people who smoke (PWS) daily, respectively; in comparison, 100% of them had serum cotinine concentrations greater than the established 10 ng/mL cut point. We compared these minor tobacco alkaloid levels to those of serum cotinine to assess their suitability as indicators of recent tobacco use at established cut points and found that their optimal cut point values would be lower than the established values. This is the first time that anabasine and anatabine are reported for urine collected from a U.S. population-representative sample of NHANES study participants, providing a snapshot of exposure levels for adults who smoked during 2013–2014. The results of this study serve as an initial reference point for future analysis of NHANES cycles, where changes in the national level of urinary anabasine and anatabine can be monitored among people who smoke to show the effect of changes in tobacco policy.

## 1. Introduction

Tobacco use is well known to be detrimental to the health of users, as well as to those around them in the case of combustible products. A tobacco-exposure biomarker measurement is an important tool for evaluating the levels of addictive or carcinogenic chemicals. Many tobacco biomarkers have been used to describe tobacco use and harmful exposure, such as nicotine and its metabolites, tobacco-specific nitrosamines, and carbon monoxide. As the levels of these chemicals increase alongside tobacco use, they can be used to estimate the frequency of tobacco use and the degree of harm associated with biomarker levels.

Several minor tobacco alkaloids such as anabasine, anatabine, nornicotine, and myosmine are structurally similar to nicotine and together make up 2–3% of the total alkaloid content of tobacco [1]. In tobacco-use biomonitoring, anabasine and anatabine are measured in the urine of people who smoke (PWS) to monitor for recent tobacco use [2]. Unlike nicotine, they have longer half-lives (~16 and ~12 h, respectively) [3]. Urinary anabasine and anatabine uniquely differentiate the use of nicotine-replacement therapy (NRT) from recent tobacco-product use [2,3]. Urinary anabasine and anatabine are also useful in identifying exposure to products containing nicotine from different sources (tobacco-derived vs. synthetic) because tobacco extracts tend to also contain anabasine and anatabine, while synthetic sources do not [4,5].

Urinary anabasine and anatabine are useful for differentiating recent tobacco use from recent NRT use because NRT products tend to contain only nicotine and no minor alkaloids; thus, measuring anabasine and anatabine together with cotinine can provide additional information than just measuring cotinine alone [6]. This differentiation also holds true for certain nicotine products that contain low or non-detectable levels of anabasine and anatabine [4]. For example, exclusive use of e-cigarettes, heat-not-burn products, or NRT products resulted in urinary anabasine and anatabine levels that were not statistically distinguishable from non-users and were significantly lower than levels resulting from exclusive cigarette smoking [7]. Additionally, numerous products are currently marketed as containing synthetic nicotine rather than tobacco-derived nicotine; measuring urinary anabasine and anatabine could differentiate people who use “tobacco-free” nicotine from those who use other tobacco-derived nicotine products [5]. Thus, measuring anabasine, anatabine, and cotinine provides useful information about the amount of nicotine absorbed as well as its source.

Biochemical verification of cigarette smoking (i.e., as a study-inclusion criterion) or abstinence (i.e., as a treatment outcome) is critical for accurately assessing harm caused by tobacco-product use. Accurate biochemical verification increases rigor and validity compared to self-reported smoking abstinence in cessation trials using NRT [8]. Cut points are used to classify people into groups based on biomarker levels. Cut points for tobacco-exposure biomarkers such as cotinine and 4-(methylnitrosamino)-1-(3-pyridyl)-1-butanol (NNAL) vary depending on demographic characteristics such as race, age, or sex [9,10]. The established cut points for urinary anabasine and anatabine are 2 ng/mL for both, and the cut point used for serum cotinine is 10 ng/mL [6,11,12]. However, additional work is needed to characterize the distribution of urinary anabasine and anatabine, as well as the efficacy of different cut points to distinguish PWS from people who do not smoke. Considering the different forms of anabasine and anatabine (free vs. total) is also crucial because a significant portion of anabasine and anatabine is secreted in a conjugated form in urine [13].

There is a lack of information on urinary anabasine and anatabine levels in the U.S. population. The National Health and Nutrition Examination Survey (NHANES) is a cross-sectional study that can be used to characterize biomarker levels that are representative of the non-institutionalized, civilian U.S. population. In this report, we analyze urinary anabasine, urinary anatabine, and serum cotinine levels within a representative sample of adults who smoke in the 2013–2014 NHANES study cycle, the first cycle where urinary anabasine and anatabine data became available. We present anabasine and anatabine levels categorized by tobacco use and demographic characteristics and compare findings for anabasine and anatabine with the gold standard tobacco-exposure biomarker (serum cotinine) for identifying recent tobacco use.

## 2. Materials and Methods

NHANES is a population-based survey designed to assess health and nutritional status through a cross-sectional observation of a complex, multistage probability-sample representative of the civilian, non-institutionalized U.S. population [14]. The survey is conducted by the National Center for Health Statistics (NCHS) of the U.S. Centers for Disease Control and Prevention (CDC). The NHANES study protocol is reviewed and approved by the NCHS Ethics Review Board and informed written consent is obtained from all participants before the study begins. Participants undergo an interview and physical examination at mobile examination centers where biological specimens are collected. Data on their responses, examination results, and laboratory measurements are made available for public use.

Urinary anabasine, urinary anatabine, and serum cotinine measurements presented in this analysis were carried out using the Urinary Nicotine Metabolites and Analogs (UNICM) method, and serum cotinine measurements were performed using the Serum Cotinine and Hydroxycotinine (SCOT) method [15,16]. The limits of detection (LOD) for urinary anabasine and anatabine were 0.51 ng/mL and 0.39 ng/mL, respectively.

PWS were defined as participants aged 18+ who self-describe having smoked at least 100 cigarettes in their lifetime and have urinary cotinine levels ≥ 20 ng/mL. We used the NHANES 2013–2014 Cotinine, Hydroxycotinine, and Other Nicotine Metabolites and Analogs-Urine-Special Sample (UCOTS_H, https://wwwn.cdc.gov/Nchs/Nhanes/2013–2014/UCOTS_H.htm (accessed on 7 April 2021)) set for this analysis. This set consisted of 2605 participants aged 18+ from a one-third sample and oversampled for people who smoke daily [17]. Participants from the special sample set were excluded from analysis if they did not respond “yes” to “smoked at least 100 cigarettes in life” (SMQ020, 1150 excluded), did not respond “every day” or “some days” to “do you now smoke cigarettes?” (SMQ040, 441 excluded), or did not have urinary cotinine levels ≥ 20 ng/mL (85 excluded). Also excluded were participants whose urinary creatinine levels were outside the range of 10–370 ng/mL (12 excluded) or who lacked data for serum cotinine (35 excluded), education (2 excluded), number of cigarettes smoked per day [(CPD), 102 excluded], or body mass index [(BMI), 8 excluded]. After exclusions, 770 participants were eligible for statistical analysis. All measurements below the limit of detection (LOD) were evaluated as the LOD divided by the square root of two.

All statistical analyses were performed using SAS 9.4 and JMP 13.2. To compare relationships between urinary anabasine, urinary anatabine, serum cotinine, and recent tobacco use, we fit weighted multiple regression models to each biomarker. We included strata and primary sampling unit (PSU) variables, and sample weights from the special sample set in all analyses to adjust for unequal probabilities of selection. Biomarker concentrations were log-transformed to adjust for skewness and heteroscedasticity. We included CPD (<0.5 pack, 0.5–<1 pack, >1 pack) in the models to represent smoking frequency. We also included demographic factors of self-reported sex, race/Hispanic origin (non-Hispanic White, non-Hispanic Black, Hispanic, Other/Multiracial), age (18–29, 30–44, 45–59, 60+), education (<High School, High School/GED, Some College, ≥Bachelor’s), and measured body mass index (Underweight/Healthy, Overweight, Obesity) in the models as covariates. For urinary anabasine and anatabine, we adjusted body-hydration levels by including urinary creatinine concentration in the models. We reported estimates and *p*-values for slopes and their 95% confidence intervals from each model. We set the statistical significance to 0.05.

## 3. Results

### 3.1. Correlation of Biomarkers

We first investigated how well these biomarkers of interest correlated with each other (Figure 1). Urinary anabasine and anatabine [ln(ng/mL)] had a strong positive correlation (R = 0.97). When correlated with serum cotinine [ln(ng/mL)], urinary anabasine and anatabine both had a moderate positive correlation (R = 0.66 and 0.68, respectively).

### 3.2. Comparison of Geometric Least Squares Means (LSM) by Demographic Characteristics

Table 1 summarizes demographic frequencies and weighted percentages of study participants by age, education, race/Hispanic origin, sex, weight status, and CPD. All point estimates passed the standards for proportions established by NCHS [18]. Participants were mainly male (53%) and non-Hispanic white (70%). Their age, education, weight, and CPD status were relatively evenly distributed across the groups. Our first analysis compared the demographic differences across urinary anabasine and anatabine to address the data gap for these biomarkers. We performed the same analysis for serum cotinine to compare the demographic influences that we found for urinary anabasine and anatabine to a well-established indicator of recent tobacco use (Appendix A). Figure 2 shows LSM levels of urinary anabasine and anatabine concentrations according to age, education, race/Hispanic origin, sex, weight status, and CPD. Overall, urinary anabasine and anatabine shared a similar pattern in comparative LSM levels among demographic subgroups, where LSM levels of anabasine were lower than anatabine for every subgroup we compared. The LSM levels in serum cotinine also showed similar patterns, especially for CPD, sex, and race.

Among all biomarkers in the CPD subgroups, the general trend was that people who smoked more cigarettes per day had higher biomarker levels than those who smoked fewer cigarettes per day. People who smoked less than half a pack per day had lower biomarker levels than those who smoked between half and less than one pack per day for urinary anabasine (*p* = 0.0024) and anatabine (*p* = 0.0019). The same was also true with even greater significance when comparing people who smoked less than half a pack per day to those who smoked one or more packs per day for urinary anabasine (*p* < 0.0001) and anatabine (*p* < 0.0001). Serum cotinine shared similar trends for the aforementioned comparisons, but we only saw differences between people who smoked half to less than one pack per day and those who smoked one or more packs per day for urinary anabasine (*p* = 0.0306) and anatabine (*p* = 0.0341).

Another demographic group that had multiple differences among subgroups was race/Hispanic origin. Non-Hispanic Blacks had higher biomarker levels than non-Hispanic Whites for urinary anatabine (*p* = 0.0286), but not for urinary anabasine (*p* = 0.0888). The same pattern held true when comparing non-Hispanic Blacks to Hispanics, where non-Hispanic Blacks had higher levels of urinary anatabine (*p* = 0.0214), but not urinary anabasine (*p* = 0.2204). Serum cotinine had a similar pattern to urinary anatabine when comparing racial/ethnic groups. As for other demographic groups, differences were only present among one or fewer groups for these biomarkers. For sex, females had higher levels than males for urinary anabasine (*p* = 0.0034) and anatabine (*p* = 0.0214). Among education groups, those who did not finish high school or earn a general education development (GED) certificate had higher urinary anabasine (*p* = 0.0126) and anatabine (*p* = 0.0424) levels than those with bachelor or advanced degrees. No differences were observed among age and weight status groups for urinary anabasine or anatabine. Overall, any significant differences among demographic subgroups for one biomarker were usually shared by the other biomarker in the same direction. Serum cotinine had similar outcomes among education and age groups, but different outcomes for sex and weight status, where no differences were observed between males and females, and people who were overweight had higher levels than those who were obese.

### 3.3. Biomarker Level Distributions

After identifying differences in biomarker levels based on CPD we looked further into biomarker level distributions among people who smoke daily (Figure 3). Anabasine and anatabine were both log-normally distributed in urine collected from people who smoke daily. Conversely, the logdistribution of cotinine in serum collected from people who smoke daily was left-skewed (Appendix B). The geometric means of people who smoke daily for urinary anabasine and anatabine were 6.28 ng/mL and 9.24 ng/mL, respectively.

We assessed the correct classification rate of people who smoke daily based on the established 2 ng/mL cut points for urinary anabasine and anatabine and found that 86% and 91%, respectively, were correctly classified (Appendix C). Conversely, people who smoke less frequently (non-daily) were less likely to be classified correctly (45% and 53% for urinary anabasine and anatabine, respectively). Only 1% of people who smoke daily had levels below the LOD for both urinary anabasine and anatabine, while 32% of people who smoke non-daily had levels below the LOD for both biomarkers. For serum cotinine, the correct classification rates for people who smoke daily and non-daily were 100% and 80%, respectively. Overall, serum cotinine had the highest correct classification rate, and the correct classification rate of urinary anatabine was slightly higher than that of urinary anabasine.

## 4. Discussion

Urinary anabasine and anatabine levels are strongly and positively correlated. These structurally similar minor alkaloids are formed in the tobacco plant as part of the same biosynthetic pathway so their levels in cigarettes and cigarette smoke would be expected to remain highly correlated [19]. Additionally, anabasine and anatabine are measured in the same matrix, so they are not differentially impacted by variable hydration and urinary dilution. Their correlations with serum cotinine are good but not as strong, which could be due to several factors such as differences in matrices and the metabolism process that cotinine undergoes but anabasine and anatabine do not. Here, we use serum cotinine as a standard of comparison as it is an excellent indicator of recent tobacco use, a trait shared with urinary anabasine and anatabine.

A primary driver for urinary anabasine and anatabine levels is CPD. We find that biomarker levels are generally higher for urinary anabasine and anatabine for people who smoke a higher number of cigarettes per day. Smoking half a pack or more per day is associated with higher biomarker levels than smoking less than half a pack per day for all biomarkers. The same trend can be seen between those who smoke between half and less than one pack per day and those who smoke one or more packs per day; this trend, however, is not observed for serum cotinine. This suggests that the relationship between CPD and the resulting tobacco exposure biomarker is strongly positively correlated but may eventually level off. A previous study reported that CPD does not accurately estimate nicotine exposure and that the reliability of using CPD as an estimator also varies by race [2]. We suspected that the same conclusion would hold true for urinary anabasine and anatabine, but their biomarker levels could have a better correlation with CPD than serum cotinine based on the significant differences we observed between higher levels of daily cigarette consumption.

Smoking frequency (non-daily, daily, low CPD, high CPD) is a major driver of urinary anabasine and anatabine levels. The geometric means of urinary anabasine and anatabine are higher in people who smoke daily compared with those who smoke non-daily. The histograms for urinary anabasine and anatabine levels of people who smoke daily are similarly shaped and have geometric means well above the cut point for recent tobacco use at 6.28 ng/mL and 9.24 ng/mL, respectively. However, the geometric means of people who smoke non-daily for urinary anabasine and anatabine are 1.41 ng/mL and 1.62 ng/mL, respectively, which are below the cut points. This demonstrates that smoking more frequently leads to higher levels of urinary anabasine and anatabine. We find similar trends in serum cotinine for biomarker levels based on smoking frequency but see slightly different distributions for people who smoke daily where urinary anabasine and anatabine have log-normal distributions while serum cotinine has a left-skewed distribution.

Demographic characteristics are associated with differences in urinary anabasine and anatabine distributions; these differences need to be considered when refining and applying cut points, as has been done for serum cotinine [3,20,21]. Among race/Hispanic origin, we find significant differences primarily between non-Hispanic Blacks and other subgroups for urinary anatabine. Non-Hispanic Blacks have higher biomarker levels than Hispanics and non-Hispanic Whites for urinary anatabine. We have seen these same differences among race/Hispanic origin for serum cotinine in previous studies; a frequently referenced reason for this observation has been differences in metabolism among racial/ethnic subgroups [22,23]. We can only speculate that similar metabolic factors may play a role in the differences we observe for non-Hispanic Blacks in urinary anatabine, but the same may not be true for urinary anabasine. For sex, we found that females have higher biomarker levels than males, which is the opposite of what is typically reported for serum cotinine, although we noted no differences between sexes for serum cotinine in this sample set [24]. Among education subgroups, we find that those who did not complete high school or receive a GED certificate have higher levels than those with a bachelor’s or other advanced degrees for all biomarkers. Even though discrepancies such as smoking prevalence and frequency exist between the upper and lower ends of education groups, those are not the reasons for the observed difference as we have corrected for CPD in the model [8]. Our findings serve as a baseline reference for the national levels of urinary anabasine and anatabine in people who smoke across multiple demographic groups. Future comparisons against this baseline can characterize changes in smoking behavior related to changes in tobacco policy.

When comparing cut point results by smoking frequency for urinary anabasine and anatabine, we find that their correct classification rates are quite similar and acceptable for identifying people who smoke daily when compared with serum cotinine. We assessed the sensitivity of cut points for urinary anabasine, urinary anatabine, and serum cotinine at their established cut points of 2 ng/mL, 2 ng/mL, and 10 ng/mL, respectively [6,11,12]. These cut points misclassified people who smoke daily at 14%, 9%, and 0% for urinary anabasine, urinary anatabine, and serum cotinine, respectively. Despite having higher misclassification rates, urinary anabasine and anatabine are useful for differentiating tobacco use from NRT product use and can complement other measures of nicotine exposure such as cotinine.

Urinary anatabine levels were higher than urinary anabasine levels. These findings are consistent with measurements of higher levels of anatabine than anabasine in most U.S. tobacco products [4]. However, the established cut points are the same (2 ng/mL) for both urinary anabasine and anatabine [6,11]. These cut points were based only on measurements of the free (unconjugated) form of these biomarkers; total measurements are preferable as they account for differences in conjugation rate [25,26]. Urinary anatabine levels could be higher than anabasine levels in total measurements but have greater similarity in free measurements because a higher proportion of anatabine may be conjugated compared with anabasine [26]. The inclusion of conjugated forms in our analysis increases the measured biomarker levels compared to free measurements alone, yet there are still PWS with urinary anabasine and anatabine levels below 2 ng/mL. Based on our findings, the optimal cut points for urinary anabasine and anatabine will likely be lower than established cut points, and the cut point for anatabine will likely be higher than anabasine in total measurements.

The study design and sample selection process limited our ability to investigate certain groups of interest. In particular, we were unable to accurately compare biomarker levels among more specific racial/ethnic subgroups such as non-Hispanic Asians or even extend our analysis to different tobacco product types such as e-cigarettes or smokeless tobacco due to the limited sample size. A moderate proportion of people who smoke non-daily were also at or below our LOD for urinary anabasine and anatabine and the sample size was small (*n* = 49), so we could not accurately assess the distribution of biomarker levels among PWS with low concentrations. Improvements in sensitivity for these biomarkers would be highly beneficial for future analyses of people who smoke infrequently.

## 5. Conclusions

We characterized urinary anabasine and anatabine distributions among a representative sample of U.S. adults who smoke and evaluated associations with demographic characteristics and smoking frequency. Urinary anabasine and anatabine are highly correlated with each other and correlate moderately with serum cotinine. By comparing the correct classification rates of the cut points among the participants in this study, we determined that urinary anabasine and anatabine have slightly lower correct classification rates for people who smoke daily than serum cotinine. We recommend that cut points be defined for total urinary anabasine and anatabine to further assist in identifying the use of tobacco-containing products. Our findings fill knowledge gaps for urinary anabasine and anatabine and demonstrate their utility as indicators of recent tobacco use when measured together with serum cotinine.

## Figures and Tables

**Figure 1 ijerph-19-09744-f001:**
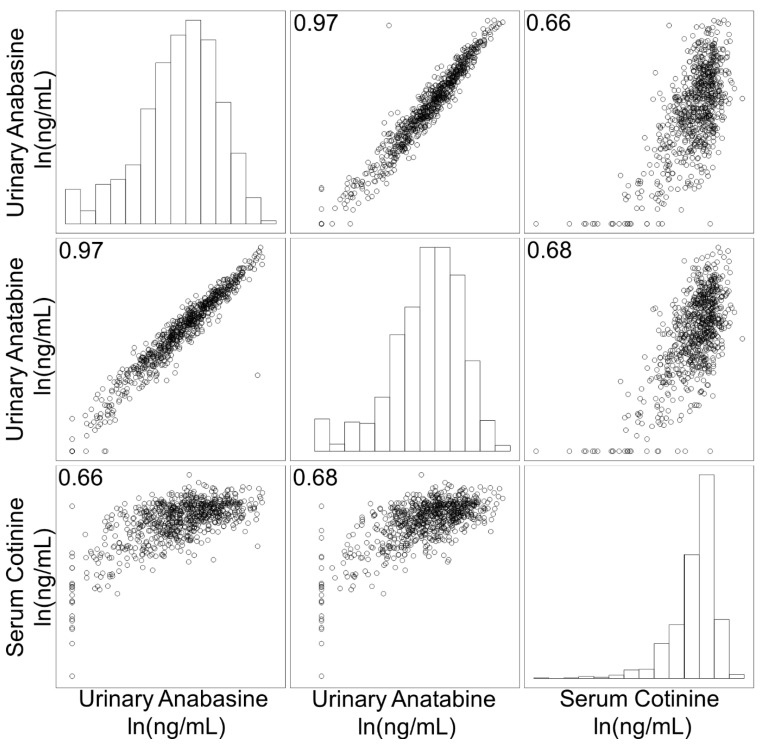
Correlation plot of the natural log of urinary anabasine, urinary anatabine, and serum cotinine levels.

**Figure 2 ijerph-19-09744-f002:**
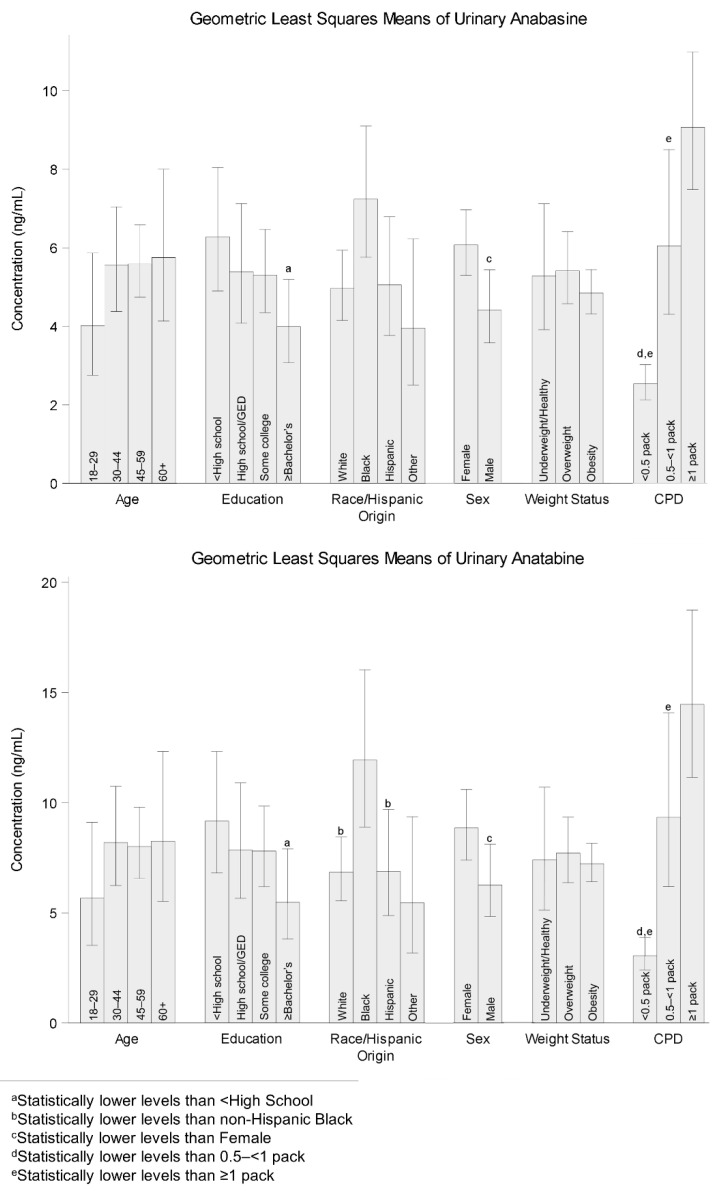
Geometric least squares means and 95% confidence intervals of urinary anabasine and anatabine by demographic characteristics. Geometric least squares means and *p*-values were calculated from natural log-transformed biomarker concentrations.

**Figure 3 ijerph-19-09744-f003:**
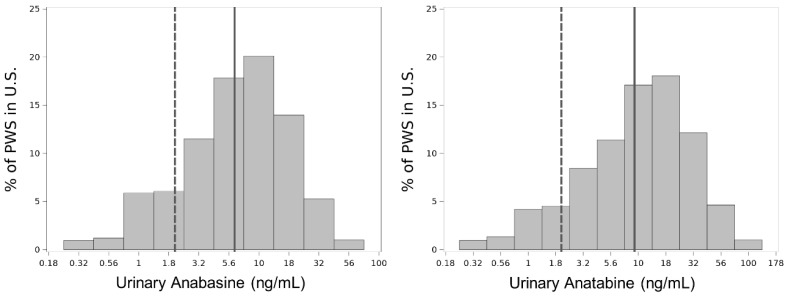
Weighted histograms of log_10_-transformed urinary anabasine and anatabine levels in people who smoke daily (*n* = 721). Frequency is presented as a percentage of people who smoke in the U.S. The presented log-transformed concentrations were exponentiated to simplify interpretation. Geometric means and cut points of each biomarker are indicated by solid and dashed gray lines, respectively.

**Table 1 ijerph-19-09744-t001:** Demographic characteristics of the participants and weighted percentages of population with the standard error (SE) of percentages.

Demographic Group	Sample Size	Percent (SE)
All	770	100
Age		
18–29	153	25.2 (1.6)
30–44	227	29.4 (2.5)
45–59	239	31.4 (3.2)
60+	151	14.0 (1.0)
Education		
<High School	234	25.7 (3.2)
High School/GED	226	29.7 (2.5)
Some College	245	32.3 (2.2)
≥Bachelor’s	65	12.3 (2.2)
Race/Hispanic Origin		
Non-Hispanic White	410	69.8 (3.1)
Non-Hispanic Black	187	14.5 (2.3)
Hispanic	99	8.6 (2.1)
Other/Multiracial	74	7.1 (1.0)
Sex		
Female	352	47.3 (2.6)
Male	418	52.7 (2.6)
Weight Status		
Underweight/Healthy	277	34.8 (2.8)
Overweight	245	32.9 (2.0)
Obesity	248	32.3 (3.6)
CPD		
1–9 (<0.5 pack)	289	37.2 (2.9)
10–19 (0.5–<1 pack)	287	35.8 (1.6)
20+ (≥1 pack)	194	27.0 (2.4)

## Data Availability

The datasets generated in this study are available in NHANES Questionnaires, Datasets and Related Documentation: UCOTS_H, COT_H [17].

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
