# Peer review of "Anabasine and Anatabine Exposure Attributable to Cigarette Smoking: National Health and Nutrition Examination Survey (NHANES) 2013–2014"

_ijerph, 2022, doi:10.3390/ijerph19159744_

Round 1

Reviewer 1 Report

Thank you for the opportunity to review “ Anabasine and Anatabine Exposure Attributable to Cigarette 2 Smoking: National Health and Nutrition Examination Survey 3 (NHANES) 2013-2014”. This manuscript   used NHANES data to examine anabasine and anatabine collected from NHANES study participants, and report exposure levels for adults who smoked during 2013-2014.  The topic is relevant; however I am concerned with a few major issues.  The paper has large room for improvement.

Major:

1.      The abstract will have to be improved, for example It is not clear in the abstract  about the purpose and meaning of the study

2.      The data seems outdated, since NHANES has released 2020 data. Can the authors please justify why the most recent data were not considered

3.      With a large proportion of subjects excluded, how much generalizability and representativeness of the results could be a concern.

4.      The authors stated in the analysis section “ We fit models for 111 self-reported sex, race/Hispanic origin (non-Hispanic White, non-Hispanic Black, His-112 panic, Other/Multiracial), age (18-29, 30-44, 45-59, 60+), education (<High School, High 113 School / GED, Some College, ≥Bachelor’s), CPD (<0.5 pack, 0.5-<1 pack, ≥1 pack), and 114 measured body mass index (Underweight/Healthy, Overweight, Obesity).” However, it is not properly described what type of models were used and the rationale of models.  The author mentioned slope, but did not describe what this slope is. 

5.      Since the data of log-transformed, the interepration of the results and p value should be based on the log-scale, for example, it is not clear if geogemtric lease square means was calculated based on log-transformed data or raw data

6.      The tables are poorly presented. The authors may want to refer to the tables in other published papers.

7.      It is not clear how the cut points were determined.

Author Response

  1. Thank you for your review of our article and the suggestion to improve the clarity of the abstract. Our primary intent for this study is to establish a reference point for the national levels of these biomarkers in people who smoke and monitoring it over time for shifts that can result from changes in smoking behavior related to changes in tobacco policy. We have modified the abstract to clarify this intent and further emphasized the implications of our results related to the previously published cut points.
  2. NHANES 2013-2014 is the first cycle where urinary anabasine and anatabine data are made available, so we want to provide this dataset as a baseline reference point for comparison in future studies. We have edited the introduction and abstract to emphasize this as our main reason for selecting this cycle. This cycle also has a special sample that oversamples for people who smoke daily to improve our sample size, which is not available in datasets after 2017.
  3. Sample size was a concern of ours for this study, and we only present data with sufficient sample size and other criteria required to be in accordance with the Standards of Proportions established by the National Center for Health Statistics. We have suppressed any variables that would not be generalizable to the U.S. population based on these criteria.
  4. We agree that the descriptions of the models, rationale, and slope were not detailed enough. We have modified the paragraph of the quoted statement to provide a more detailed overview of our statistical analysis.
  5. We have calculated the geometric least squares means using log-transformed data and interpreted the results and p-values based on the log-scale as well. We have added this clarification in the captions of Figure 2 and Figure A1 where these values are presented.
  6. Table 1 and Table A1 have been revised to reduce unnecessary columns and rows so that the design is more intuitive and consistent with other published tables while maintaining the format requested by the journal.
  7. The cut point values used in this article were not determined by our analysis and were established by reference articles. We compared our results to them to assess their sensitivity, but we agree that this is confusing because it is not stated that we obtained these values from outside references until the discussion. We have moved this statement to the introduction as it is more appropriate and less confusing to establish cut point values in that section.

Reviewer 2 Report

I have read the article entitled "Anabasine and Anatabine Exposure Attributable to Cigarette Smoking: National Health and Nutrition Examination Survey (NHANES) 2013-2014". This is an interesting study on a specific topic. The article is well written and the figures used support the presentation of the results. However, it is worthwhile for the authors to consider extending the introduction section, which is quite short compared to the other sections. I also encourage you to verify the percentages in the tables.

Author Response

  1. Thank you for your review of our article. We agree that the introduction is comparatively shorter and have moved content from the discussion that would be more appropriately placed in the introduction.
  2. We have added a decimal place to the percentages in Table 1 to remediate any issues caused by rounding to whole numbers. The percentages for each demographic category should now add up to 100%.

Reviewer 3 Report

This study assessed the relationship between urinary anabasine and anatabine. These two compounds could also help identify tobacco products use. Overall, this article is interesting and written well. I have a few concerns as follows.

1   1.  My first question is why authors chose to use data year 2013-2014 rather than the most recent year or a relevant later year? I think using the old year data is out of date.

     2.  Figure 1 should be clearer; it is hard to read the values on X/Y-coordinates.

     3.  I would suggest reframing the content in discussion. For example, the first sentence/paragraph could go to the introduction part.

     4. What are the implications of the findings for this study? Discussion should expand this. For example, differentiate the nicotine sources (tobacco-derived vs. synthetic) is an advantage, while why we need this? How will this contribute to future clinical trials design as well as inform policies? And how to monitor compliance with smoking cessation programs?

Author Response

  1. Thank you for your review of our article. NHANES 2013-2014 is the first cycle where urinary anabasine and anatabine data are made available, so we want to use this dataset as a baseline reference point for comparison in future studies. We have rephrased the abstract and introduction to emphasize our reasoning for selecting this cycle. This cycle also has a special sample that oversamples for people who smoke daily to improve our sample size, which is not available in datasets after 2017.
  2. Thank you for pointing out the image quality for Figure 1. We had shrunk it down for the sake of formatting, but at a loss of resolution. We have changed the font and increased the resolution of the figure so that it is easier to read.
  3. We agree that the first paragraph in the discussion would be more appropriate in the introduction as it mostly covers background information from our references rather than the results of our analysis. We have moved this as suggested.
  4. This study characterizes the U.S. population-weighted baseline for urinary anabasine and anatabine among people who smoke. However, the tobacco product marketplace is becoming more diverse, and many people are using products that result in nicotine uptake with less smoke exposure. Thus, the measurement of nicotine metabolites such as cotinine is best complimented by also measuring other biomarkers such as anabasine and anatabine. Our current paper provides a first step in this direction. Our objective is to show the national exposure levels of these biomarkers and then monitor how the national levels shift over time in response to changes in tobacco policy. We have modified the abstract, introduction, discussion, and conclusion sections to further emphasize this as the primary purpose of the article. As our analysis does not present results that support the featured biomarkers’ ability to differentiate nicotine sources, we believe it would be more appropriate to reserve the discussion of these implications in an article with results that support this application.

Round 2

Reviewer 1 Report

Thank you for addressing my comments. The manuscript has been improved.  

Table 1 can be further revised to report sample size ( weighted proportion). No need to report the standard error. 

Reviewer 3 Report

I believe this manuscript is much improved and I think it is ready to be accepted. Just do proof reading and check before being published. Thanks.